# A Commercial Probiotic Induces Tolerogenic and Reduces Pathogenic Responses in Experimental Autoimmune Encephalomyelitis

**DOI:** 10.3390/cells9040906

**Published:** 2020-04-07

**Authors:** Laura Calvo-Barreiro, Herena Eixarch, Manuel Ponce-Alonso, Mireia Castillo, Rafael Lebrón-Galán, Leyre Mestre, Carmen Guaza, Diego Clemente, Rosa del Campo, Xavier Montalban, Carmen Espejo

**Affiliations:** 1Servei de Neurologia-Neuroimmunologia, Centre d’Esclerosi Múltiple de Catalunya, Vall d’Hebron Institut de Recerca, Hospital Universitari Vall d’Hebron, Passeig Vall d’Hebron 119-129, 08035 Barcelona, Spain; laura.calvo@vhir.org (L.C.-B.); herena.eixarch@vhir.org (H.E.); micastillo@vhebron.net (M.C.); xavier.montalban@cem-cat.org (X.M.); 2Universitat Autònoma de Barcelona, 08193 Bellaterra, Cerdanyola del Vallès, Spain; 3Red Española de Esclerosis Múltiple (REEM), Fondo de Investigación Sanitaria, Instituto de Salud Carlos III, Ministerio de Economía y Competitividad, 28801 Madrid, Spain; rlebron@sescam.jccm.es (R.L.-G.); leyre@cajal.csic.es (L.M.); cgjb@cajal.csic.es (C.G.); dclemente@sescam.jccm.es (D.C.); 4Servicio de Microbiología, Hospital Universitario Ramón y Cajal, and Instituto Ramón y Cajal de Investigación Sanitaria (IRYCIS), Carretera de Colmenar km. 9.1, 28034 Madrid, Spain; lugonauta@gmail.com (M.P.-A.); rosacampo@yahoo.com (R.d.C.); 5Grupo de Neuroinmuno-Reparación, Unidad de Investigación, Hospital Nacional de Parapléjicos, Finca “La Peraleda” s/n, 45071 Toledo, Spain; 6Grupo de Neuroinmunología, Departamento de Neurobiología Funcional y de Sistemas, Instituto Cajal, CSIC, Avenida Doctor Arce 37, 28002 Madrid, Spain; 7Division of Neurology, University of Toronto, St. Michael’s Hospital, 30 Bond Street, Toronto, ON M5B 1W8, Canada

**Keywords:** gut microbiota, probiotics, immune regulation, experimental autoimmune encephalomyelitis, multiple sclerosis, adaptive immunity, antigen presenting cells, gut microbiome, gut permeability

## Abstract

Previous studies in experimental autoimmune encephalomyelitis (EAE) models have shown that some probiotic bacteria beneficially impact the development of this experimental disease. Here, we tested the therapeutic effect of two commercial multispecies probiotics—Lactibiane iki and Vivomixx—on the clinical outcome of established EAE. Lactibiane iki improves EAE clinical outcome in a dose-dependent manner and decreases central nervous system (CNS) demyelination and inflammation. This clinical improvement is related to the inhibition of pro-inflammatory and the stimulation of immunoregulatory mechanisms in the periphery. Moreover, both probiotics modulate the number and phenotype of dendritic cells (DCs). Specifically, Lactibiane iki promotes an immature, tolerogenic phenotype of DCs that can directly induce immune tolerance in the periphery, while Vivomixx decreases the percentage of DCs expressing co-stimulatory molecules. Finally, gut microbiome analysis reveals an altered microbiome composition related to clinical condition and disease progression. This is the first preclinical assay that demonstrates that a commercial probiotic performs a beneficial and dose-dependent effect in EAE mice and one of the few that demonstrates a therapeutic effect once the experimental disease is established. Because this probiotic is already available for clinical trials, further studies are being planned to explore its therapeutic potential in multiple sclerosis patients.

## 1. Introduction

Multiple sclerosis (MS) is a chronic, degenerative autoimmune disease and the most common inflammatory demyelinating disorder of the central nervous system (CNS) worldwide [1]. Although the aetiology of the disease is still unclear, MS could be triggered in the periphery by activated autoreactive immune cells that subsequently infiltrate into the CNS or by some CNS-intrinsic events [1]. Both innate and adaptive immune responses participate in MS pathogenesis. Thus, pro-inflammatory autoreactive T helper (Th) 1 and Th17 populations are the main pathological CD4^+^ T cells implicated in this disease. Moreover, both the presence of autoreactive CD8^+^ T cells in the CNS lesions and oligoclonal bands in the cerebrospinal fluid of MS patients and the pathogenic effect of autoantibodies on myelin sheaths highlight the key role of both CD8^+^ T and B cells in this disease [2]. However, the well-known defective function of regulatory T (T_reg_) cells in MS patients [3] can also partially explain the disease pathogenesis, since peripheral tolerance mechanisms are essential to prevent the self-reactivity of circulating autoreactive immune cells. Given this relationship, some therapeutic approaches try to skew pro-inflammatory responses towards enhancing anti-inflammatory (e.g., Th2 cells) or even immunoregulatory (e.g., T_reg_ cells) populations to suppress autoreactive populations.

Recently, the commensal microbiota has emerged as a putative environmental risk factor for MS. Studies in experimental autoimmune encephalomyelitis (EAE) models have shown that the commensal microbiota is an essential player in triggering autoimmune demyelination [4,5]. In fact, the lack of microbial stimuli in germ-free or antibiotic-treated mice compared with specific pathogen-free animals resulted in decreased demyelination and cell infiltration levels in the CNS and, consequently, lower disease severity during the clinical course of EAE [5,6]. However, experimental data support the idea that some bacterial strains, far from being harmful, have a beneficial impact on the outcome of EAE. Thus, the promotion of beneficial microorganisms via probiotics is being developed as an important therapeutic strategy involving the gut microbiota in EAE [7,8,9,10,11,12,13].

In the present study, we investigated the therapeutic impact of two commercially available probiotics—Lactibiane iki and Vivomixx—composed by different strains from bacteria genera *Lactobacillus, Bifidobacterium*, and *Streptococcus*, on the clinical outcome of established EAE. Lactibiane iki contains two probiotic strains that have previously proved their capacity of increasing immunoregulatory cytokine interleukin (IL)-10 in vitro and of diminishing clinical severity in experimental colitis [14]. Furthermore, Vivomixx treatment has also demonstrated to induce IL-10 in a mouse model of colitis [15] as well as to promote anti-inflammatory immune responses in experimental diabetes [16]. Recently, Weiner laboratory has described the anti-inflammatory effect of Vivomixx treatment in MS patients that seems to be related to monocyte and dendritic cell (DC) functions [17,18]. Our results suggest that Lactibiane iki plays a noticeable role in both the immune response and CNS inflammation and demyelination in this experimental model of MS, being capable of reverting already established clinical signs. Therefore, this probiotic could exert beneficial effects in MS patients and could be rapidly translated into the clinic since it is already a commercialized product.

## 2. Materials and Methods

### 2.1. Mice

C57BL/6JOlaHsd 8-week-old female mice purchased from Envigo (Venray, The Netherlands) were used. Mice were housed under standard light- and climate-controlled conditions, and standard chow and water were provided ad libitum. All experiments were performed in strict accordance with EU (Directive 2010/63/UE) and Spanish regulations (Real Decreto 53/2013; Generalitat de Catalunya Decret 214/97). The Ethics Committee on Animal Experimentation of the Vall d’Hebron Research Institute approved all procedures described in this study (protocol number: 35/15 CEEA).

### 2.2. Induction and Assessment of EAE

Anaesthetized mice were immunized by subcutaneous injection of 100 μl of phosphate-buffered saline containing 200 μg of peptide 35–55 from myelin oligodendrocyte glycoprotein (MOG_35–55_, Proteomics Section, Universitat Pompeu Fabra, Barcelona, Spain) emulsified in 100 μl of complete Freund’s adjuvant (incomplete Freund’s adjuvant (IFA, F5506, Sigma-Aldrich, St- Louis, MO, USA) containing 4 mg/ml *Mycobacterium tuberculosis* H37RA (231141, BD Biosciences, San Jose, CA, USA)). At 0 and 2 days postimmunization (dpi), mice were intravenously injected with 250 ng of pertussis toxin (P7208, Sigma-Aldrich). Mice were weighed and examined daily for neurological signs in a blinded manner using the following criteria: 0, no clinical signs; 0.5, partial loss of tail tonus for 2 consecutive days; 1, paralysis of whole tail; 2, mild paraparesis of one or both hind limbs; 2.5, severe paraparesis or paraplegia; 3, mild tetraparesis; 4, tetraparesis (severe in hind limbs); 4.5, severe tetraparesis; 5, tetraplegia; and 6, death [19]. Corrective measures and endpoint criteria to ensure EAE-incident animals welfare included (i) wet food pellets in the bed-cage to facilitate access to food as well as hydration, (ii) subcutaneous administration of 0.5 ml of glucosaline serum (glucose 10%) in case of more than 15% of weight loss, and (iii) mouse euthanasia if the weight loss exceeds 30% or an animal reaches the clinical score of 5. All data presented are in accordance with the guidelines suggested for EAE publications [20] and with the ARRIVE (Animal Research: Reporting of In Vivo Experiments) guidelines for animal research.

### 2.3. Motor Function Assessment

At 33 dpi, motor performance was evaluated using a Rotarod apparatus (Ugo Basile, Gemonio, Italy) that was set to accelerate from a speed of 4 to 40 rotations per minute in a 300-second time trial. Each mouse was given four trials. Once mice were placed on the rotating cylinder, the amount of time that the animals walked on the cylinder without falling was recorded.

### 2.4. Bacterial Strains and Treatments

Lactibiane iki (Pileje, Paris, France) is composed of *Bifidobacterium lactis* LA 304, *Lactobacillus acidophilus* LA 201, and *Lactobacillus salivarius* LA 302. Vivomixx (Grifols, Barcelona, Spain) is composed of *Lactobacillus acidophilus* (DSM 24735), *Lactobacillus plantarum* (DSM 24730), *Lactobacillus paracasei* (DSM 24733), *Lactobacillus delbrueckii subsp. bulgaricus* (DSM 24734), *Bifidobacterium longum* (DSM 24736), *Bifidobacterium breve* (DSM 24732), *Bifidobacterium infantis* (DSM 24737), and *Streptococcus thermophilus* (DSM 24731). Before therapeutic administration with a single or a double daily dose and after attaining a clinical score equal to or greater than 2 or 1, between 13 and 16 dpi or 12 and 15 dpi, respectively, mice were randomized into clinically equivalent experimental groups. Administration of a 200-μl volume containing 1.6 × 10^9^ colony-forming units (CFU) of Lactibiane iki, 9 × 10^9^ CFU of Vivomixx, or water (vehicle) via oral gavage was performed once or twice daily, depending on experimental conditions, until the end of the experiment (34 dpi). The selection of the proper probiotic dosage was done keeping previous preclinical assays in mind [9,10] but also was limited by probiotic solubility rates. 

### 2.5. Ex Vivo Splenocyte Proliferative Capacity

Splenocyte suspensions were prepared by grinding spleens of EAE mice through a 70-μm nylon cell strainer at 34 dpi. Splenocytes were seeded at 2 × 10^5^ cells per well in X-VIVO^TM^ 15 medium (BE02-060F, Lonza, Basel, Switzerland) supplemented with 1% *v/v* L-glutamine (X0550, Biowest, Nuaillé, France), 0.4% *v/v* penicillin-streptomycin (L0022, Biowest), 0.1 M HEPES (H0887, Sigma-Aldrich), and 6 μM 2-β-mercaptoethanol (M3148, Sigma-Aldrich) within 96-well plates. Splenocyte cultures were stimulated with 5 μg/mL MOG_35–55_ or 5 μg/mL phytohaemagglutinin-L (PHA-L, L2769, Sigma-Aldrich) and compared to non-stimulated (control) condition. After 54 h in vitro, 75 μL of supernatant were harvested and stored at −80 °C to further assess cytokine secretion. At the same time, cell cultures were again supplemented with completed medium containing 1 μCi of [^3^H]-thymidine per well (NET027Z, PerkinElmer, Waltham, MA, USA). Splenocyte cultures were maintained under the same conditions for an additional 18 h, and incorporated radioactivity was measured in a beta-scintillation counter (Wallac, Turku, Finland). Five replicates per condition (control, MOG_35–55_, and PHA-L) and mouse were analysed and the results are shown as the stimulation index. Stimulation indices were calculated by dividing the mean counts per minute (cpm) of MOG_35–55_ or PHA-L condition by the mean cpm of the control condition.

### 2.6. Cytokine Detection by Luminex

Cytokine secretion (granulocyte-macrophage colony-stimulating factor (GM-CSF); interferon gamma (IFN-γ), IL-2, IL-4, IL-6, IL-10, IL-12p70, IL-17A, IL-21, IL-22, and IL-23; and tumour necrosis factor alpha (TNF-α)) was assessed in the supernatants of stimulated splenocytes by using a ProcartaPlex Multiplex Immunoassay (Invitrogen, Carlsbad, CA, USA), according to the manufacturer’s instructions. Data were analysed with a Magpix instrument (Luminex Corporation, Austin, TX, USA) and ProcartaPlex Analyst software (Thermo Fisher Scientific, Waltham, MA, USA).

### 2.7. RNA Isolation, cDNA Synthesis, and qRT-PCR

At 34 dpi, spinal cords of euthanized EAE mice were collected, immersed in liquid nitrogen, and stored at −80 °C until use. Total RNA was extracted from spinal cords using TRI Reagent (T9424, Sigma-Aldrich) and pretreated with TURBO DNase (AM1907, Invitrogen) in order to remove any genomic DNA trace. Next, mRNA was reverse transcribed with a High-Capacity cDNA Reverse Transcription Kit with RNase Inhibitor (4368814, Applied Biosystems, Foster City, CA, USA). Primers for *Tbx21* (Mm00450960), *Gata3* (Mm00484683), *Rorγt* (custom assay as described in Reference [21]), *Foxp3* (Mm00475162), *Ifng* (Mm01168134), *Il4* (Mm00445259), *Il17a* (Mm00439618), *Il10* (Mm01288386), *Tgfb1* (Mm01178820), *Mrc1* (Mm01329362), and the housekeeping gene *Gapdh* (Mm99999915) as well as TaqMan Gene Expression Master Mix (4369016, all from Applied Biosystems) were used to perform qPCR according to the manufacturer’s instructions. The relative level of gene expression was calculated using the 2^−ΔΔCT^ method [22]. Briefly, the expression of the housekeeping gene (Gapdh) was used for normalization and the expression of the genes of interest (Tbx21, Gata3, Rorγt, Foxp3, Ifnγ, Il4, Il17a, Il10, Tgfb1, and Mrc1) in the vehicle-treated condition was used as a calibrator. No template control (NTC), no reverse transcriptase control (NRT), and no amplification control (NAC) samples were included in qPCR experiments. Analyses were performed with SDS 2.4 software (Applied Biosystems), and any sample with a quantification cycle value of greater than 35 was considered a non-amplified sample [23].

### 2.8. Flow Cytometry Analysis

Spleen cell suspensions were prepared as described previously. Cell subsets were analysed using fluorochrome-conjugated monoclonal antibodies (mAbs) after discrimination of dead cells by Fixable Viability Stain (BD Pharmingen, BD Biosciences). For analysis of the T_reg_ cell population, CD3ε (553061, BD Pharmingen), CD4 (561090, BD Pharmingen), and CD25 (12-0251, eBioscience, San Diego, CA, USA) were used. FoxP3 intracellular staining was performed using fluorochrome-labelled anti-FoxP3 mAb (17-5773, eBioscience). CD39 (25-0391, eBioscience), CD62L (563252, BD Horizon, BD Biosciences), Helios (563801, BD Pharmingen), and ICOS (CD278, 564592, BD Horizon) were also selected and evaluated in the T_reg_ cell population. For DC subpopulations and activation status, mAbs specific for B220 (553088, BD Pharmingen), CD11b (562605, BD Horizon), CD11c (553802, BD Pharmingen); CD8a (47-0081, eBioscience), and CD317 (17-3172, eBioscience); CD80 (553769, BD Pharmingen); CD86 (563077, BD Pharmingen); major histocompatibility complex class II (MHCII, 562363, BD Pharmingen); and PD-L1 (CD274, 124315, Biolegend, San Diego, CA, USA) were used. For analysis of macrophage, neutrophil, and myeloid-derived suppressor cell (MDSC) populations, CD11b (562605, BD Horizon), CD206 (141712, Biolegend), F4/80 (12-4801, eBioscience), Ly6C (560593, BD Pharmingen), and Ly6G (551460, BD Pharmingen) were used; for B cell subsets, B220 (552772, BD Pharmingen), CD1d (553846, BD Pharmingen), CD5 (550035, BD Pharmingen), CD19 (560143, BD Pharmingen), CD138 (562610, BD Horizon), and MHCII (562363, BD Pharmingen) were used. For analysis of T cell activation status, CD3ε (553062, BD Pharmingen), CD4 (46-0042, eBioscience), CD8a (563152, BD Horizon), CTLA-4 (CD152, 564331, BD Pharmingen), LAG-3 (CD223, 552380, BD Pharmingen), PD-1 (CD279, 135217, Biolegend), and TIM-3 (CD366, 25-5870, eBioscience) were selected and assessed. For intracellular cytokine determination, ex vivo stimulation of splenocytes was performed with 50 ng/ml phorbol 12-myristate 13-acetate (PMA, P1585, Sigma-Aldrich) and 1 μg/ml ionomycin (I0634, Sigma-Aldrich) in the presence of GolgiPlug and GolgiStop (555029 and 554724, BD Biosciences) for 6 h. Then, CD3ε (553061, BD Pharmingen), CD4 (46-0042, eBioscience), and CD8a (563152, BD Horizon) staining was performed. Cytokine intracellular staining was performed using fluorochrome-labelled anti-IFN-γ (563376, BD Horizon), anti-IL-4 (554436, BD Pharmingen), anti-IL-10 (563276, BD Horizon), and anti-IL-17A (561020, BD Pharmingen) mAbs and anti-CD69 mAb (560689, BD Pharmingen) to assess proper ex vivo stimulation. Fluorescence was analysed with a CytoFLEX flow cytometer and CytExpert 2.3 software (Beckman Coulter, Brea, CA, USA).

### 2.9. Histological Analysis

At 34 dpi, spinal cords of euthanized EAE mice were collected, fixed in a 4% paraformaldehyde solution, embedded in paraffin, and cut into 4-μm thick coronal sections. Demyelination was assessed by rabbit polyclonal anti-myelin basic protein (MBP, AB980, Merck, Kenilworth, NJ, USA); CD3^+^ infiltrating cells was assessed by rabbit polyclonal anti-CD3 (A0452, Dako, Agilent Technologies, Santa Clara, CA, USA); axonal damage was assessed by mouse monoclonal purified anti-neurofilament H, nonphosphorylated antibody (SMI32, 801701, Biolegend); and reactive microglia and astroglia were assessed by lectin from *Lycopersicon esculentum* (LEA, L0651, Sigma-Aldrich) and mouse monoclonal anti-glial fibrillary acidic protein (GFAP, C9205, Sigma-Aldrich), respectively. Briefly, coronal spinal cord sections were deparaffinized and rehydrated, and antigen retrieval (when needed) and preincubation in blocking solution for 1 h at room temperature were performed. Immunofluorescences were carried out by incubating sections with the corresponding primary antibody diluted in the blocking solution overnight at 4 °C. After rinsing, sections were incubated with the corresponding streptavidin-conjugated fluorochrome or fluorescent secondary antibody in the blocking solution for 1 h at room temperature when needed. Finally, cell nuclei were stained with 4’,6-diamidino-2-phenylindole (DAPI, D9542, Sigma-Aldrich) and coverslips were mounted with Fluoromount-G (00-4958-02, Invitrogen).

Images were acquired using a Leica AF6000 fluorescence microscope and Las AF visualization software (Leica Microsystems, Wetzlar, Germany), and mosaic images were obtained at a magnification of 20× and analysed using ImageJ. For every staining condition, two mosaic images from the thoracic spinal cord, each separated by 100–200 μm, were selected per mouse and evaluated in a blinded manner. For demyelination measurements, the results are shown as the percentage of white matter area without MBP staining relative to the total white matter area. For analysis of inflammation, the total number of CD3^+^ cells within the infiltrated CNS tissue was assessed by manually counting cells. The density of stained cells was considered in relation to the whole white matter area. For quantification of axonal damage and of microglia and astrocyte reactivity, the area with specific staining relative to the total white matter area was analysed using an ImageJ-adapted *macro*. Briefly, after selection of the whole white matter area, a threshold for immunofluorescence was assessed for each marker and fluorescence was measured, with the results given in μm^2^. The results are shown as the percentage of positive antigen-specific area relative to the total white matter area.

### 2.10. In Vivo Intestinal Permeability Studies

At 34 dpi, EAE mice were weighed and orally gavaged with an isotonic solution of 0.9% NaCl with fluorescein sodium salt (NaF, F6377, Sigma-Aldrich) at 10 μg/g mouse body weight or without NaF (negative control mice). After 1 h, mice were euthanized and blood samples were collected in heparinized tubes. NaF concentration in plasma was measured in flat bottom 96-well plates (Nunc, Roskilde, Denmark) by spectrophotofluorimetry with a 485-nm excitation wavelength and 535-nm emission wavelength in a Thermo Scientific Appliskan (Thermo Fisher Scientific) as previously described [24]. NaF standard concentrations were used as reference, and both samples and calibration curve were performed in duplicate.

### 2.11. Stool Sample Collection, DNA Extraction, Library Preparation, and 16S rDNA Sequencing

Faeces were freshly collected in duplicate from representative mice at −1 dpi (n = 12, untreated naïve mice) and 12 dpi (n = 12, untreated EAE mice), considering the cage, clinical score, and cumulative score when appropriate. Faecal samples were also freshly gathered in duplicate from every treated EAE mouse (n = 8 per group of treatment: Lactibiane iki, Vivomixx or vehicle) at 33 dpi. After collection, samples were frozen by immersion in liquid nitrogen and stored at −80 °C.

Total DNA was extracted from each sample with a QiaAMP extraction kit (Qiagen, Hilden, Germany). Then, the V3-V4 region of the bacterial 16S rDNA gene was amplified and the PCR products were pooled equally by following the 16S Metagenomic Sequencing Library Preparation guidelines (Illumina, San Diego, CA, USA). Finally, 16S rDNA massive sequencing was performed on a MiSeq (Illumina) platform using paired-end, 300-base reads at FISABIO (Valencia, Spain). In total, 48 faecal samples and a negative control sample were sequenced and subjected to microbiome analysis. Raw sequence data of faecal samples were deposited in GenBank (BioProject ID: PRJNA545034).

### 2.12. Microbiome Bioinformatics

Bioinformatics analysis was performed using the Quantitative Insights Into Microbial Ecology version 2 (QIIME2) software suite (version 2019.1) [25]. Raw sequence data (FASTQ files) were demultiplexed and quality filtered using the q2-demux plugin and were subsequently denoised and merged with the DADA2 pipeline [26] (via the q2-dada2 plugin) to identify all amplicon sequence variants (ASVs) [27] and their relative abundance in each sample. To minimize the number of spurious ASVs, unique sequences with a total abundance of less than 7 reads across all samples were filtered out [28]. ASVs were first aligned and then used to construct a phylogenetic tree via the align-to-tree-mafft-fasttree pipeline [29,30] in the q2-phylogeny plugin. ASVs were taxonomically classified by using the classify-sklearn naïve Bayes taxonomy classifier (via the q2-feature-classifier plugin) [31] against the Silva 132 99% operational taxonomic units (OTUs) reference database [32]. Sequences not assigned to any taxa (unassigned) or classified as chloroplast, mitochondria, or eukaryote sequences were discarded. The taxonomic profiles of samples were visualized using the q2-taxa plugin. Diversity analysis was performed using the q2-diversity plugin, and samples were then rarefied (subsampled without replacement) to 56,390 sequences per sample. We selected this rarefaction depth since it guaranteed robust diversity measures and retained all samples according to the rarefaction plot. The diversity analysis comprised both alpha diversity metrics (Shannon index and Faith’s Phylogenetic Diversity (Faith-pd) index [33], which measure microbiome richness) and beta diversity metrics (unweighted UniFrac [34] and weighted UniFrac [35], which measure differences in the microbiome composition while up-weighting differences in ASV phylogenetic distances). Unweighted UniFrac reports differences in the presence or absence of ASVs, while weighted UniFrac reports differences in the presence, absence, and abundance of ASVs.

### 2.13. Statistical Analysis

All comparisons in the clinical, histological, immunological, gene expression, and intestinal permeability studies were performed using the differences of least-squares means. A normal distribution was assumed for all studied variables except for CD3, which showed a better fit with a lognormal distribution. When repeated measures within mice were performed, compound symmetry was used as the covariance structure. When only a potential clustering effect of the experiments was present, a variance components structure showed an acceptable fit. All analyses were carried out with the Proc MIXED program, except for analysis of CD3, for which Proc GLIMMIX was used. All tests were two-tailed, and statistical significance was set at a *p* value of <0.05.

Regarding the microbiome statistical analysis, the differences in mean alpha diversity metrics were calculated by a Kruskal–Wallis test [36]. To test for differences in the microbiome composition between groups, we performed principal coordinate analysis (PCoA) based on the beta diversity unweighted UniFrac and weighted UniFrac distance metrics. Permutational multivariate analysis of variance (PERMANOVA) [37] was performed to determine which categorical variable factors explained statistically significant variances in the microbiota composition, whereas a Mantel test [38] was used for continuous variables. All statistical tests were conducted via the q2-diversity plugin of QIIME2. To determine which specific taxa explained beta diversity differences or just relative differences between groups, differential abundance analyses were performed in variables that yielded statistically significant differences in beta diversity analysis and in variables that did not yield statistically significant differences in beta diversity, but they did have a biological value. For categorical variables, the linear discriminant analysis (LDA) effect size (LEfSe) method was used for testing taxonomic comparisons [39]. LEfSe combines the standard tests for statistical significance (Kruskal–Wallis test and pairwise Wilcoxon test) with linear discriminate analysis for taxa selection. Besides detecting significant features, it also ranks features by effect size, which put features that explain most of the biological difference on top. The alpha value for the factorial Kruskal–Wallis test was 0.05, and the threshold for the logarithmic LDA score for discriminative taxa was set at 2.0. However, assessment of the differential taxa abundance for continuous variables was performed using Gneiss [40] via the q2-gneiss plugin of QIIME2. Gneiss constructs taxa balances and performs multivariate response linear regression in order to assess if any of those balances shows statistically significant differences along value distribution of the response variable of interest. Finally, the log ratios of abundance values for selected taxa within samples were plotted using the q2-deicode and q2-qurro plugins in QIIME2 [41,42].

## 3. Results

### 3.1. Lactibiane iki Improves the EAE Clinical Outcome in a Dose-Dependent Manner as a Therapeutic Approach

Mice were treated once daily with Lactibiane iki, Vivomixx, or vehicle after attaining a clinical score equal to or greater than 2 and were randomized from 13–16 dpi to the end of the experiment (34 dpi). Whereas treatment with Lactibiane iki improved the clinical outcome (area under the curve (AUC): 72.60 ± 18.12, n = 20, *p* = 0.029) compared to vehicle treatment (AUC: 83.88 ± 8.48, n = 18), Vivomixx treatment did not ameliorated the clinical course of already established EAE (AUC: 75.35 ± 18.73, n = 20, *p* = 0.100) (Figure 1a,b). Mice were weighed daily to monitor their well-being (Figure 1c), and the Rotarod test was performed to assess motor coordination skills at 33 dpi. No differences were observed regarding weight loss, but compared to vehicle treatment (15.82 sec ± 9.62 sec, n = 18), both Lactibiane iki (25.75 sec ± 17.54 sec, n = 20, *p* = 0.040) and Vivomixx (27.64 sec ± 20.96 sec, n = 20, *p* = 0.035) treatment improved motor coordination skills (Figure 1d).

To investigate the dose-response effect, we doubled the administered dosage of probiotics by treating mice twice daily. We observed greater clinical improvement in mice treated with Lactibiane iki (AUC: 62.78 ± 23.63, n = 17, *p* = 0.008) than in vehicle-treated mice (AUC: 80.57 ± 10.82, n = 17), but still, Vivomixx treatment did not reach statistical significance (AUC: 69.47 ± 20.49, n = 17, *p* = 0.057) (Figure 1e,f).

### 3.2. Pathogenic Responses are Reduced in the Spinal Cords of Probiotic-Treated EAE Mice

Since our treatments improved motor function skills and Lactibiane iki specifically ameliorated clinical score, we aimed to investigate the histological signs under clinical improvement. Within the spinal cord white matter, the percentage of demyelination was lower in mice treated with Lactibiane iki (6.25% ± 2.91%, n = 8, *p* = 0.003) or Vivomixx (6.46% ± 2.55%, n = 9, *p* = 0.002) than in vehicle-treated mice (14.27% ± 5.64%, n = 9) (Figure 2a), and T cell inflammatory infiltrate density was lower in mice treated with Lactibiane iki (70.11 cells/mm^2^ ± 58.43 cells/mm^2^, n = 8, *p* = 0.041) or Vivomixx (71.68 cells/mm^2^ ± 74.47 cells/mm^2^, n = 9, *p* = 0.023) than in vehicle-treated mice (199.65 cells/mm^2^ ± 164.27 cells/mm^2^, n = 9) (Figure 2b). In addition, the level of axonal damage tended to decrease in mice treated with Lactibiane iki (0.90% ± 0.59%, n = 8, *p* = 0.071) compared to vehicle-treated mice (1.44% ± 0.55%, n = 9) (Figure 2c), although these differences were not statistically significant. No differences were observed in microglia and astrocyte reactivity between groups (*data not shown*).

Characteristic transcription factors and cytokines related to EAE pathogenesis were also analysed in the spinal cords at 34 dpi. Although no significant differences were observed for the majority of the studied genes (Appendix A) and several genes were even not detected (*Il17a*, *Il4*, and *Il10*), a 4-fold decrease in the expression of the Th17-defining transcription factor *Rorγt* was observed in mice treated with Lactibiane iki (2^−∆∆Ct^: 0.60 ± 0.36, n = 8, *p* = 0.016) compared to that in vehicle-treated mice (2^−∆∆Ct^: 1.09 ± 0.48, n = 9) (Figure 3a).

### 3.3. Lactibiane iki Reduces Antigen-Specific Proliferation but Does Not Modify Disease-Related Cytokine Profile

A reduction in antigen-specific proliferation was observed under Lactibiane iki treatment compared to that in the vehicle group (Stimulation index: 4.38 ± 1.43, n = 12 vs. 6.07 ± 2.36, n = 12, respectively; *p* = 0.045) (Figure 3b). However, no differences were detected in antigen-specific proliferation for Vivomixx treatment (Figure 3b) or in the polyclonal immune response for any treatment (Figure 3c). Moreover, no between-group differences in the cytokine secretion pattern were observed for any ex vivo stimulatory condition (Appendix A). Finally, when pro-inflammatory (IFN-γ and IL-17A), anti-inflammatory (IL-4), or immunoregulatory (IL-10) intracellular cytokine patterns were studied in T cell population, no differences were observed in Lactibiane iki- or Vivomixx-treated mice compared to vehicle group (*data not shown*).

### 3.4. Lactibiane iki Increases T_reg_ Cells and Diminishes Plasma Cells in the Periphery

The induction of immunoregulatory responses is involved in the resolution of inflammation and can potentially alleviate clinical symptoms. Thus, the increase in the T_reg_ cell (CD3^+^CD4^+^CD25^+^FoxP3^+^) population in mice treated with Lactibiane iki (9.05% ± 2.14%, n = 7, *p* = 0.009) compared to vehicle-treated mice (6.02% ± 1.11%, n = 6) could be related to the observed clinical improvement (Figure 4a). However, no differences were observed between the experimental groups in the levels of the cell markers CD39, CD62L, HELIOS, and ICOS in T_reg_ cells (*data not shown*). Regarding B lymphocytes, Lactibiane iki decreased the percentage of peripheral plasma cells (B220^−^CD19^−^MHCII^−^CD138^+^: 0.83% ± 0.24%, n = 8, *p* = 0.007) when compared to vehicle-treated mice (1.23% ± 0.27%, n = 8) whereas Vivomixx-treated mice (1.00% ± 0.18%, n = 8, *p* = 0.059) only tended to lower this pro-inflammatory cell population (Figure 4b).

### 3.5. Commercial Probiotics Modulate the Number and Phenotype of Antigen Presenting Cells (APCs)

Both Lactibiane iki (11.80% ± 3.74%, n = 9, *p* = 0.001) and Vivomixx (9.28% ± 2.75%, n = 9, *p* = 0.017) treatment increased the percentage of myeloid DCs (mDCs: B220^−^CD11b^+^CD11c^high^CD8a^−^) in the periphery compared to that in the vehicle group (6.26% ± 2.02%, n = 9) (Figure 5a). Several studies have demonstrated that CNS mDC-like cells, which are part of the CNS infiltrate and are well positioned to activate myelin-specific T cells in both EAE and MS [43,44], are derived from haematopoietic cells that infiltrate the CNS [43]. Thus, we further investigated the specific mDC phenotype in the periphery that could be related to clinical improvement in the mice. Under Lactibiane iki treatment, mDCs presented an immature or semi-mature phenotype (lower expression of the maturation marker MHCII (*p* = 0.034, n = 9) and a trend to CD80 reduction (*p* = 0.079, n = 9) (Figure 5b,c)), and an increased number of PD-L1-expressing mDCs (*p* = 0.001, n = 9) was observed when compared to vehicle mice (Figure 5d). On the other hand, a lower percentage of mDCs expressing the co-stimulatory molecule CD86 was observed in mice treated with Vivomixx (10.83% ± 1.51%, n = 9, *p* = 0.049) than in vehicle-treated mice (12.60% ± 1.99%, n = 9) (Figure 5e). Further study of the T cell activation status (CTLA-4, LAG-3, PD-1, and TIM-3) and other myeloid populations (total macrophages, M2 macrophages and monocytic MDSCs) did not show differences between groups (*data not shown*). However, neutrophils/granulocytic MDSCs (CD11b^+^Ly6G^+^Ly6C^low^) were reduced in Lactibiane iki-treated mice (56.44% ± 4.56%, n = 9, *p* = 0.004) when compared to vehicle-treated group (66.06% ± 7.43%, n = 9) (Figure 5f).

### 3.6. Commercial Probiotics do not Alter Intestinal Permeability but do Modify Gut Microbiome Composition

As mice treated with Lactibiane iki showed a decrease in autoreactive cell proliferation (Figure 3b) and circulating MOG-reactive T cells have been described to induce pathological changes in intestinal morphology and function [24], intestinal permeability was evaluated at 34 dpi. However, no differences were observed regarding this pathogenic sign in Lactibiane iki-treated mice (3939 ng NaF/ml ± 1653 ng NaF/ml, n = 8, *p* = 0.147) or in Vivomixx-treated mice (6403 ng NaF/ml ± 2871 ng NaF/ml, n = 8) when compared to vehicle-treated mice (5818 ng NaF/ml ± 3036 ng NaF/ml, n = 8) (Figure 6a).

Regarding microbiome profiling, 16S rDNA sequencing of total faecal samples (n = 48: n = 12 untreated naïve mice (−1 dpi); n = 12 untreated EAE mice (12 dpi); n = 8 vehicle-treated mice (33 dpi); n = 8 Lactibiane iki-treated mice (33 dpi); and n = 8 Vivomixx-treated mice (33 dpi)) yielded 9,382,015 reads, and the negative control yielded 53 reads, which eliminated the possibility of high contamination levels. After quality control, quality filtering, merging, chimaera removal, and exclusion of very low frequency sequences, a total of 5,346,132 reads were ultimately obtained. After taxonomic assignment, unassigned and eukaryotic sequences were removed, and 5,345,803 reads of 784 ASVs were ultimately used in downstream analyses.

Two different alpha diversity indices, Shannon and Faith-pd, were analysed in the comparisons of interest: a) *treatment*: vehicle- vs. Lactibiane iki- vs. Vivomixx-treated mice, and b) *clinical condition*: untreated naïve mice (−1 dpi) vs. untreated EAE mice (12 dpi) vs. vehicle-treated mice (33 dpi). Regarding treatment effect, no significant differences were found at 33 dpi. However, when clinical condition was evaluated, the Shannon index was increased in EAE mice, 12 dpi (6.24 ± 0.36, n = 12) compared with that in either naïve (5.43 ± 0.41, n = 12, *p* < 0.001) or EAE mice, 33 dpi (5.57 ± 1.12, n = 8, *p* = 0.037) (Figure 6b), but the Faith-pd index did not differ.

Regarding beta diversity studies, no significant differences were observed between treatments at 33 dpi. However, clinical condition did change the overall microbial community structure according to both unweighted UniFrac (naïve −1 dpi vs. EAE 12 dpi, *p* = 0.001; naïve −1 dpi vs. EAE 33 dpi, *p* = 0.003; EAE 12 dpi vs. EAE 33 dpi, *p* = 0.005) and weighted UniFrac (naïve −1 dpi vs. EAE 12 dpi, *p* = 0.001; naïve −1 dpi vs. EAE 33 dpi, *p* = 0.002; EAE 12 dpi vs. EAE 33 dpi, *p* = 0.001) methods (Figure 6c). Besides, we were interested in studying beta diversity along *disease progression*, which can be measured by the quantitative variable accumulated score (the sum of daily clinical score per mouse) between every EAE mice included in the experiment (n = 36: n = 12 untreated EAE mice (12 dpi); n = 8 vehicle-treated mice (33 dpi); n = 8 Lactibiane iki-treated mice (33 dpi); and n = 8 Vivomixx-treated mice (33 dpi)). Thus, disease progression also changed the overall microbial community structure according to both the unweighted UniFrac (*p* = 0.003) and weighted UniFrac (*p* = 0.007) methods (Figure 6d).

Although no differences were observed regarding beta diversity, we sought to determine whether the relative abundance of specific taxonomic groups differed among treatments at 33 dpi. Lactibiane iki caused an increase in the relative abundance of several taxa, including the genera *Lachnoclostridium* and *Bifidobacterium*, whereas Vivomixx enhanced different microbial taxa, including *Streptococcus* (Appendix A). Regarding clinical condition, EAE mice at 12 dpi exhibited a higher abundance of several taxa belonging to the orders *Clostridiales* and *Bacteroidales* (class *Clostridia*), while EAE mice at 33 dpi exhibited an increase in bacteria belonging to the orders *Lactobacillales* (class *Bacilli*) and *Clostridiales* (class *Clostridia*) (Appendix A).

### 3.7. Specific Bacterial Taxa are Associated with Disease Progression

The quantitative variable accumulated score, which is related to *disease progression* and had previously exhibited differences in beta diversity (Figure 6d), was studied by the Gneiss method [40]. This statistical method establishes multiple balance trees for taxa and performs multivariate regression to reveal whether any of those trees present significant differences relative to the distribution of values of the studied variable (accumulated score). Two different balance trees were selected due to their relevance and significant difference: y0 (*false discovery rate (FDR)-corrected p value* < 0.001)) and y9 (*FDR-corrected p value* < 0.001). Whereas the class *Bacteroidia* and the genus *Enterococcus* were overrepresented in the numerator of both the y0 and y9 balance trees (related to a higher accumulated score in mice), the families *Lachnospiraceae* and *Atopobiaceae* and the genus *Bifidobacterium* were highly represented in the denominator of the y0 and y9 balance trees (related to a lower accumulated score in mice), respectively (Figure 7a,b). Subsequent visual inspection of the previously mentioned taxa corroborated the higher abundance of both the family *Atopobiaceae* and the genus *Bifidobacterium* in mice with a lower accumulated score (Figure 7c). However, no such tendency was observed within the *Lachnospiraceae* family (Figure 7c). Finally, visual inspection of *Enterococcus* revealed a higher abundance in mice with a higher accumulated score, while no differences were observed regarding the class *Bacteroidia* (Figure 7c).

## 4. Discussion

Our study is the first preclinical assay that uses commercial multispecies probiotics—Lactibiane iki and Vivomixx—in a therapeutic manner as a translational approach that would accelerate their availability to patients. It is also the first study to demonstrate a dose-dependent effect of probiotic treatment in an EAE model and one of the few therapeutic approaches that demonstrate a clinical effect once the experimental disease is established (mice randomization after attaining, at least, a mild paraparesis of hind limbs). Thus, we show that oral treatment with Lactibiane iki improves the clinical outcome of EAE mice in a dose-dependent manner as a therapeutic approach. The clinical improvement was related to decreased CNS demyelination and inflammation, as corroborated by many previous studies of probiotic treatment [7,8,9,10,11,12,13]. Lactibiane iki treatment also decreases the expression of the Th17-defining transcription factor *Rorγt* in the spinal cord, revealing a reduction in this pro-inflammatory cell population previously connected to both EAE [45] and MS [46] pathogenesis in the CNS. In fact, Th17 cells can directly contact neurons, establish immune-neuronal synapses without T-cell receptor engagement, and transect neural axons [47]. Thus, the observed trend towards a reduction in axonal damage under Lactibiane iki treatment could be partially explained by a decrease in Th17 cell–neuronal interaction.

We show that EAE improvement is associated with an increase in the frequency of T_reg_ cells and with a reduction in the frequency of plasma cells in mice under Lactibiane iki treatment. However, no significant change in the frequency of T_reg_ or plasma cells was observed under Vivomixx treatment, consistent with prior studies in MS patients [17,18]. T_reg_ cells can suppress pathogenic immune responses by reducing or modulating the population of effector T cells mediated through immunosuppressive cytokine secretion or cell–cell interaction [48,49,50]. T_reg_ cells can also modulate several types of cells, including DCs, and can directly suppress autoantibody-producing plasma cells, among others [51,52]. As evidence supports the pathogenic role of demyelinating antibodies synthesized outside the CNS in MS [53], the decreased frequency of peripheral plasma cells could be related to both decreased demyelination and clinical improvement. Similarly, Lactibiane iki treatment successfully limits the encephalitogenic immune response in the periphery. Thus, we hypothesize that gut exposure to probiotic could reduce autoreactive responses by promoting T_reg_ cells, as previously described [7,8,9,10,11].

Lactibiane iki treatment increases the number and modifies the phenotype of mDCs towards an immature or semi-mature profile. Semi-mature DCs were previously described to induce tolerance through the secretion of immunosuppressive cytokines (e.g., IL-10 and TGF-β), the expression of surface markers (e.g., PD-L1 and PD-L2), and the promotion of T_reg_ cells [54]. In fact, Lactibiane iki-treated mice exhibit a higher number of PD-L1-expressing DCs and promotion of T_reg_ cells. The previously mentioned molecules have been characterized as markers for tolerogenic DCs (_tol_DCs) [54], but only PD-L1 exhibits a contact-dependent mechanism for modulating peripheral immune responses and tolerance induction [55,56]. PD-1/PD-L1 interaction has been described as a key event in several autoimmune diseases (*reviewed in* Reference [57]), including MS and EAE [56,58,59]. In fact, PD-1/PD-L1 signalling limits pro-inflammatory responses through the modulation and maintenance of T_reg_ cells, promotion of CD8^+^ T cell tolerance, and restriction of self-reactive T cells during antigen presentation by DCs [60,61]. Moreover, bidirectional communication between DCs and T_reg_ cells could partially explain the induction of _tol_DCs, since T_reg_ cells can signal back to DCs and promote their differentiation towards a tolerogenic phenotype [62]. Similarly, the promotion of _tol_DCs, which can present a wide range of epitopes to effector cells, could extend tolerance to multiple antigen specificities [62]. Thus, the induction of _tol_DCs and T_reg_ cells by Lactibiane iki could be related to the promotion of immune tolerance and the restriction of self-reactive T cells and pro-inflammatory responses in the periphery. Regarding Vivomixx treatment, a decrease in the population of mDCs expressing the co-stimulatory molecule CD86, which is required for suitable T cell stimulation by APCs, would indicate an inefficient T cell activation profile. Finally, Lactibiane iki also reduced the neutrophil/granulocytic MDSC population in the periphery. However, as these two immune cell populations share morphology and cell surface markers and no immune suppression studies were performed [63], we cannot claim which population was affected by probiotic administration.

Intestinal permeability is a pathological hallmark associated with EAE [24] and MS [64] that potentially supports disease progression. Previously, circulating MOG-reactive T cells have been described to induce pathological changes in intestinal morphology and to function as soon as 7 dpi [24]. We initially thought that the decrease in autoreactive cell proliferation under Lactibiane iki treatment could be associated to a lower intestinal permeability and, as a consequence, a lower pro-inflammatory intestinal environment. However, no statistical significance was found between Lactibiane iki- or Vivomixx- and vehicle-treated mice at the end of the experiment. On the contrary, a reduction of intestinal permeability has previously been showed in the early phases (7 and 14 dpi) of EAE mice treated with probiotics in a prophylactic approach [12]. This suggests that, once intestinal features are established, the ability to revert this altered gut permeability in a therapeutic approach is limited.

We also found that the clinical condition changes the global microbial community, a change explained partially by the increased abundance of taxa in the orders *Clostridiales* and *Bacteroidales* in the acute phase and in the orders *Lactobacillales* and *Clostridiales* in the chronic phase of EAE. The families *Ruminococcaceae* and *Lachnospiraceae* (order *Clostridiales*) and the family *Bacteroidaceae* (order *Bacteroidales*) have been described as highly prevalent in MS patients [65] and, together with taxa belonging to the family *Rikenellaceae* (order *Bacteroidales*), as also dominant in healthy individuals [66]. Members of these families have been previously associated with butyrate production, which is highly relevant because it promotes T_reg_ cell differentiation and activity and ultimately suppresses pro-inflammatory responses [67,68]. Thus, overrepresentation of these bacteria would be an attempt to compensate for the excessive pro-inflammatory immune responses due to the experimental disease. 

Although no differences are observed regarding beta diversity between treatments, the administration of Lactibiane iki, composed by genera *Lactobacillus* and *Bifidobacterium*, is associated with an increased abundance of the genus *Lachnoclostridium* (family *Lachnospiraceae*) and several taxa belonging to the family *Bifidobacteriaceae*, being this latter taxon consistent with probiotic composition itself. As previously mentioned, the family *Lachnospiraceae* has been previously associated with butyrate production and has also been correlated with IL-10 and TGF-β production by different immune cells [65]. Regarding *Bifidobacteria*, probiotics composed of different strains including *Bifidobacterium* have exhibited beneficial effects in EAE [11,69] and MS [17,18] and have been correlated with anti-inflammatory immune markers. Concerning Vivomixx, composed by genera *Lactobacillus, Bifidobacterium*, and *Streptococcus*, the observed increase in the genus *Streptococcus* (family *Streptococcaceae*) is consistent with probiotic composition, with prior studies in Vivomixx-treated MS patients, and connected to anti-inflammatory responses affecting mainly DCs and monocytes [17,18].

Although disease progression changes the overall microbial community structure, subsequent visual inspection did not reveal an association between every discovered taxonomic group and this parameter. Only specific subordinate taxa belonging to the discovered taxonomic groups revealed associations with EAE progression. Interestingly, Lactibiane iki-treated mice exhibited a higher abundance of *Bifidobacterium* than mice in the other treatment groups, which was associated with a lower clinical score. Finally, vehicle treatment was correlated with a higher abundance of *Enterococcus*, connected to higher clinical scores. However, information regarding the effect of *Enterococcus* on the host immune system is scarce, except that this genus is widely described to contain several pathogenic bacteria although some members are used as probiotics due to their capacity to secrete bacteriocins and prevent diarrhoea [70].

## 5. Conclusions

Our data show that the commercial probiotic Lactibiane iki improved the clinical outcome of EAE mice in a dose-dependent manner as a therapeutic approach whereas Vivomixx did not perform a beneficial clinical effect in the experimental disease. This clinical improvement was related to decreased demyelination and T cell infiltration in the CNS potentially caused by diminished pro-inflammatory and increased immunoregulatory immune responses in the periphery. On the other hand, administration of either probiotic modulated the number and phenotype of APCs, specifically, DCs. Finally, we described that both the clinical condition and disease progression alter the gut microbiome composition. Our results emphasize that Lactibiane iki plays a noticeable role in the immune response and in the processes of CNS demyelination and inflammation in this EAE model, being capable of reverting already established clinical signs. Because this probiotic is already available for clinical trials, further studies are being planned to explore its therapeutic potential in MS patients.

## Figures and Tables

**Figure 1 cells-09-00906-f001:**
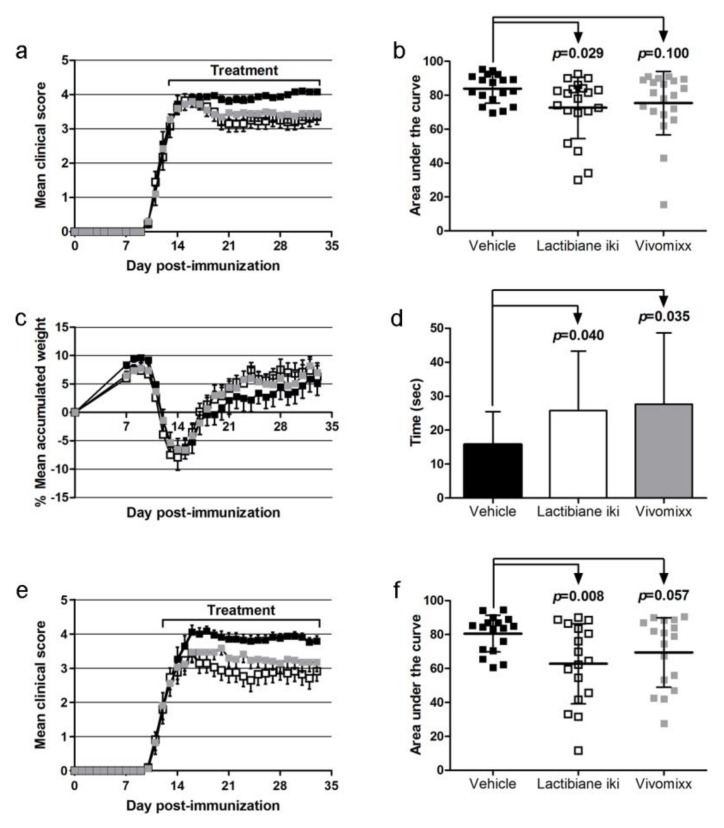
Therapeutic administration of commercial probiotic Lactibiane iki improves the clinical outcome in a dose-dependent manner in mice with established experimental autoimmune encephalomyelitis (EAE). C57BL/6JOlaHsd mice were immunized by subcutaneous injection of peptide 35–55 from myelin oligodendrocyte glycoprotein (MOG_35–55_) and weighed and examined daily for neurological signs. After attaining a clinical score equal to or greater than 2 and being randomized (13–16 days postimmunization (dpi)), mice were treated once daily with Lactibiane iki (1.6 × 10^9^ colony-forming units (CFU)/dose), Vivomixx (9 × 10^9^ CFU/dose), or vehicle (water); weighed; and examined daily for neurological signs in a blinded manner until the end of the experiment (34 dpi). Daily oral therapeutic treatment with a single dose of Lactibiane iki improved the clinical score (**a**) and area under the curve (AUC) (**b**) of treated mice, whereas Vivomixx treatment did not ameliorate disease course (**a**,**b**). On the other hand, none of the probiotics affected weight loss (**c**). Motor coordination skills were evaluated using a Rotarod apparatus at 33 dpi. Both Lactibiane iki and Vivomixx treatment improved motor function skills (**d**). The charts (**a**–**d**) present the combined results of three independent experiments (vehicle, n = 18; Lactibiane iki, n = 20; and Vivomixx, n = 20). On the other hand, when the previously mentioned oral doses of commercial probiotics were administered twice daily, a greater improvement in the clinical score (**e**) and AUC (**f**) of Lactibiane iki-treated mice were observed. However, no statistical significance was observed in the Vivomixx group. The graphs (**e**,**f**) present the combined results of two independent experiments (vehicle, n = 17; Lactibiane iki, n = 17; and Vivomixx, n = 17). The data are presented as the means ± standard errors of the mean (**a**,**c**,**e**) or the means ± standard deviations (**b**,**d**,**f**). ■, Vehicle; □, Lactibiane iki; ■, Vivomixx.

**Figure 2 cells-09-00906-f002:**
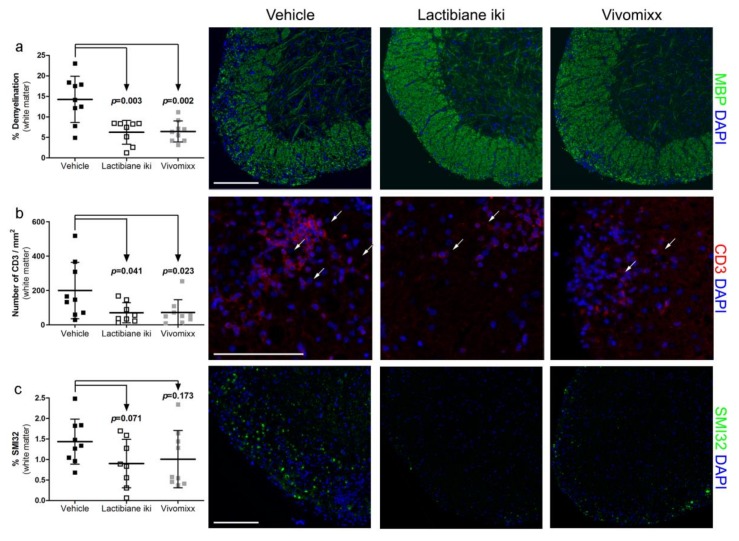
Probiotic treatments ameliorate histopathological hallmarks in the central nervous system (CNS) of experimental autoimmune encephalomyelitis (EAE) mice. At 34 days postimmunization (dpi), spinal cords of euthanized EAE mice were collected, fixed in a 4% paraformaldehyde solution, embedded in paraffin, and cut into 4-μm thick coronal sections. Demyelination was assessed by anti-myelin basic protein (MBP) antibody; CD3^+^ infiltrating cells were assessed by anti-CD3 antibody; axonal damage was assessed by anti-neurofilament H, nonphosphorylated (SMI32) antibody; and reactive microglia and astroglia were assessed by lectin from *Lycopersicon esculentum* and anti-glial fibrillary acidic protein antibody, respectively. For demyelination measurements, the results are shown as the percentage of white matter area without MBP staining relative to the total white matter area. For analysis of inflammation, the total number of CD3^+^ cells within the infiltrated CNS tissue was assessed by manually counting cells. The density of stained cells was considered in relation to the whole white matter area. For quantification of axonal damage and of microglia and astrocyte reactivity, the area with specific staining relative to the total white matter area was analysed. Daily therapeutic administration of commercial probiotics Lactibiane iki and Vivomixx reduced demyelination (**a**) and T cell infiltrates (**b**). However, only Lactibiane iki treatment tended to decrease axonal damage (**c**) although statistical significance was not reached. No differences were observed in microglia and astrocyte reactivity (*data not shown*). The graphs show the results from representative mice (vehicle, n = 9; Lactibiane iki, n = 8; and Vivomixx, n = 9) in two independent experiments under double dose administration. Arrows indicate CD3^+^ cells. Scale bars indicate 100 µm. The data are presented as the means ± standard deviations. Abbreviations: DAPI: 4’,6-diamidino-2-phenylindole; MBP: myelin basic protein; SMI32: anti-neurofilament H, nonphosphorylated.

**Figure 3 cells-09-00906-f003:**
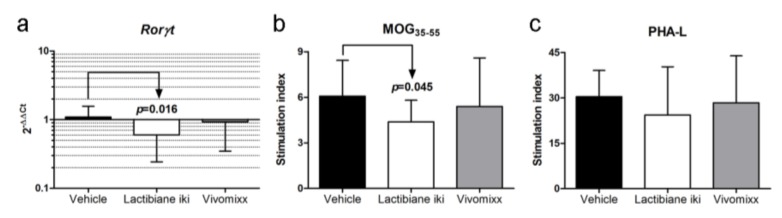
Lactibiane iki diminishes pathogenic responses in the central nervous system (CNS) and antigen-specific response in the periphery. Total RNA was extracted from spinal cords of euthanized experimental autoimmune encephalomyelitis (EAE) mice at 34 days postimmunization (dpi). Next, genomic DNA trace was removed, and mRNA was reverse transcribed. Primers for *Tbx21*, *Gata3*, *Rorγt*, *Foxp3*, *Ifng*, *Il4*, *Il17a*, *Il10*, *Tgfb1*, and *Mrc1* and the housekeeping gene *Gapdh* were selected as key pathogenic genes of EAE, and their relative level of gene expression was calculated using the 2^−ΔΔCT^ method. Spinal cords from Lactibiane iki-treated mice experienced a 4-fold decrease in the expression of the Th17-defining transcription factor *Rorγt* when compared to vehicle group (**a**). The graph shows the results of a representative experiment under double dose administration (vehicle, n = 9; Lactibiane iki, n = 8; and Vivomixx, n = 8). On the other hand, splenocyte suspensions were prepared by grinding spleens at 34 dpi. Lactibiane iki also reduced the proliferative capacity of antigen-specific immune cells in response to peptide 35–55 from myelin oligodendrocyte glycoprotein (MOG_35–55_) stimulation (**b**), but none of the multispecies probiotics alter the polyclonal response (**c**). The graphs present the combined results of two independent experiments under single dose administration (vehicle, n = 12; Lactibiane iki, n = 12; and Vivomixx, n = 13). The data are presented as the means ± standard deviations. Abbreviations: MOG_35—55_: peptide 35–55 from myelin oligodendrocyte glycoprotein; PHA-L: phytohaemagglutinin-L; *Rorγt*: RAR-related orphan receptor gamma t.

**Figure 4 cells-09-00906-f004:**
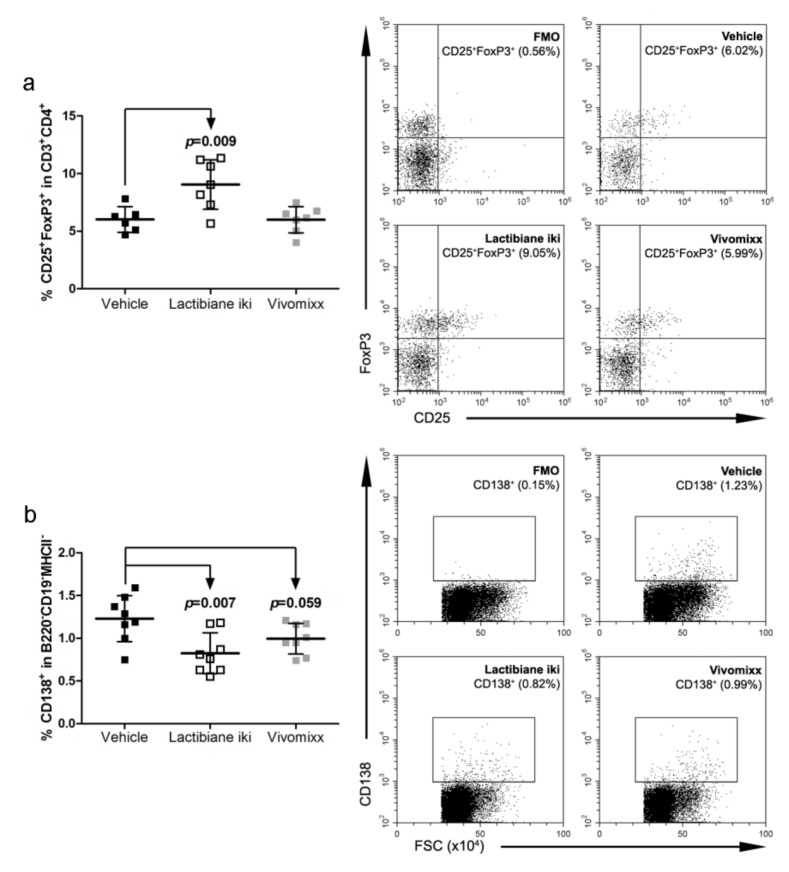
Lactibiane iki increases regulatory T (T_reg_) cells and diminishes plasma cells in the periphery. Splenocyte suspensions were prepared by grinding spleens of experimental autoimmune encephalomyelitis (EAE) mice through a 70-μm nylon cell strainer at 34 days postimmunization (dpi). Cell subsets were analysed after exclusion of doublets and dead cells from the gating scheme. Lactibiane iki increased the T_reg_ cell population (CD3^+^CD4^+^CD25^+^FoxP3^+^) in the periphery when compared to vehicle group (**a**). The graph presents the results of a representative experiment under single dose administration (vehicle, n = 6; Lactibiane iki, n = 7; and Vivomixx, n = 7). On the other hand, Lactibiane iki treatment reduced plasma cells (B220^−^CD19^−^MHCII^−^CD138^+^) in the periphery when compared to vehicle-treated mice (**b**). The graph presents the results of a representative experiment under double dose administration (vehicle, n = 8; Lactibiane iki, n = 8; and Vivomixx, n = 8). Representative flow cytometry dot plots of FMO (fluorescence minus one) control and experimental groups are shown for every population-defining gate. The data are presented as the means ± standard deviations. Abbreviations: FMO: fluorescence minus one; FSC: forward scatter.

**Figure 5 cells-09-00906-f005:**
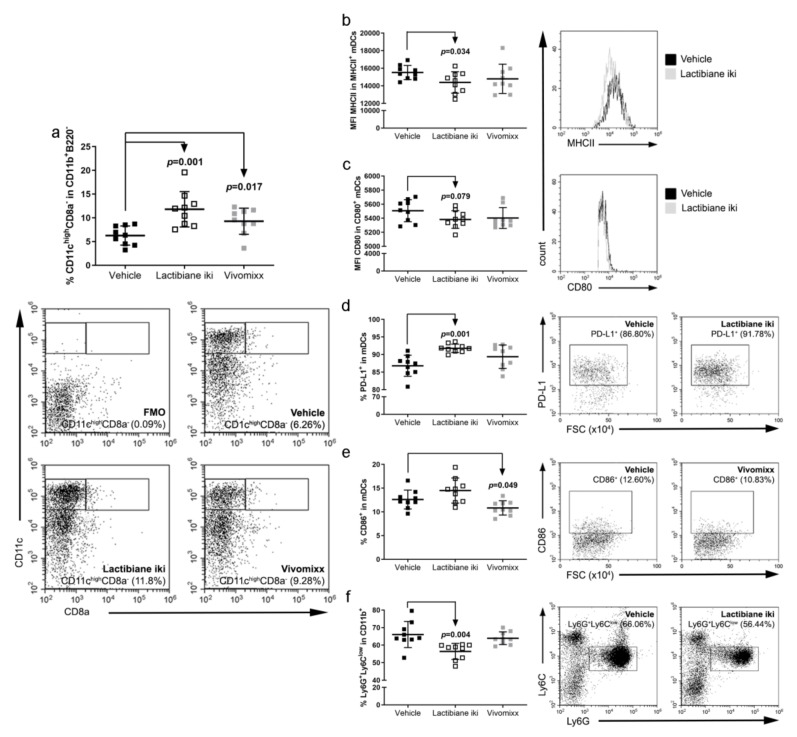
Commercial probiotics modulate the number and phenotype of antigen presenting cells (APCs). Splenocyte suspensions were prepared by grinding spleens of experimental autoimmune encephalomyelitis (EAE) mice through a 70-μm nylon cell strainer at 34 days postimmunization (dpi). Cell subsets were analysed after exclusion of doublets and dead cells from the gating scheme. Administration of either probiotic increased the population of myeloid dendritic cells (mDCs: CD11b^+^B220^−^CD11c^high^CD8a^−^) in the periphery (**a**). Lactibiane iki decreased the expression of major histocompatibility complex class II (MHCII) (**b**) and CD80 (**c**) on the surface of mDCs. In addition, Lactibiane iki increased the PD-L1^+^ population of mDCs (**d**), whereas Vivomixx reduced the population of mDCs expressing the co-stimulatory molecule CD86 (**e**). Lactibiane iki also decreased the neutrophils/granulocytic myeloid-derived suppressor cell (MDSC) population in the periphery (**f**). The graphs present the results of a representative experiment under double dose administration (vehicle, n = 9; Lactibiane iki, n = 9; and Vivomixx, n = 9). Representative flow cytometry dot plots or histograms are shown (**a**–**f**). The data are presented as the means ± standard deviations. Abbreviations: FMO: fluorescence minus one; FSC: forward scatter; mDCs: myeloid dendritic cells; MFI: median fluorescence intensity.

**Figure 6 cells-09-00906-f006:**
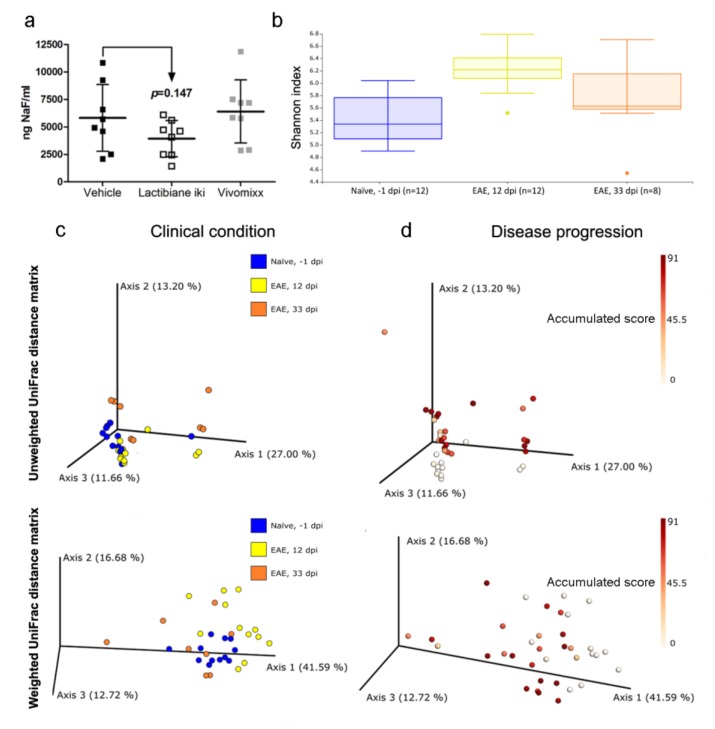
Commercial probiotics do not alter intestinal permeability, and microbial populations are modified by clinical condition and disease progression. At 34 days postimmunization (dpi), experimental autoimmune encephalomyelitis (EAE) animals were weighed and orally gavaged with an isotonic solution of fluorescein sodium salt (NaF) at 10 μg/g mouse body weight. After 1 h, mice were euthanized, and plasma samples were collected. Intestinal permeability was evaluated by measuring the NaF concentration in mouse plasma. None of the probiotics reduced intestinal permeability compared to vehicle group (**a**). The graph presents the results of a representative experiment under double dose administration (vehicle, n = 8; Lactibiane iki, n = 8; and Vivomixx, n = 8). Faeces were freshly collected in duplicate from representative mice at −1 dpi (n = 12, untreated naïve mice) and 12 dpi (n = 12, untreated EAE mice), considering the cage, clinical score, and cumulative score when appropriate. Faecal samples were also freshly gathered in duplicate from every treated EAE mouse (n = 8 per group of treatment: vehicle, Lactibiane iki or Vivomixx) at 33 dpi. After collection, samples were frozen by immersion in liquid nitrogen and stored at −80 °C. Once 16S rDNA sequencing and quality controls of total faecal samples (n = 48: n = 12 untreated naïve mice (−1 dpi); n = 12 untreated EAE mice (12 dpi); n = 8 vehicle-treated mice (33 dpi); n = 8 Lactibiane iki-treated mice (33 dpi); and n = 8 Vivomixx-treated mice (33 dpi)) were performed, alpha diversity was analysed. The alpha diversity as measured by the Shannon index was altered when clinical condition was analysed (untreated naïve mice (−1 dpi) vs. untreated EAE mice (12 dpi) vs. vehicle-treated mice (33 dpi)). Shannon index increased after EAE induction in the acute phase (12 dpi) but returns to the level seen in naïve mice in the chronic phase of EAE (33 dpi) (**b**). On the other hand, beta diversity was also studied. Both the clinical condition (**c**) and disease progression (measured by the quantitative variable *accumulated score*) (**d**) alter the beta diversity as measured by both the unweighted and weighted UniFrac. The graphs represent clinical condition analysis (n = 32: n = 12 untreated naïve mice (−1 dpi); n = 12 untreated EAE mice (12 dpi); and n = 8 vehicle-treated mice (33 dpi)) and disease progression analysis (n = 36: n = 12 untreated EAE mice (12 dpi); n = 8 vehicle-treated mice (33 dpi); n = 8 Lactibiane iki-treated mice (33 dpi); and n = 8 Vivomixx-treated mice (33 dpi)) of a representative experiment under double dose administration. The data are presented as the means ± standard deviations in the (**a**) graph and as a box plot (**b**). Abbreviations: dpi: days postimmunization; EAE: experimental autoimmune encephalomyelitis; NaF: fluorescein sodium salt.

**Figure 7 cells-09-00906-f007:**
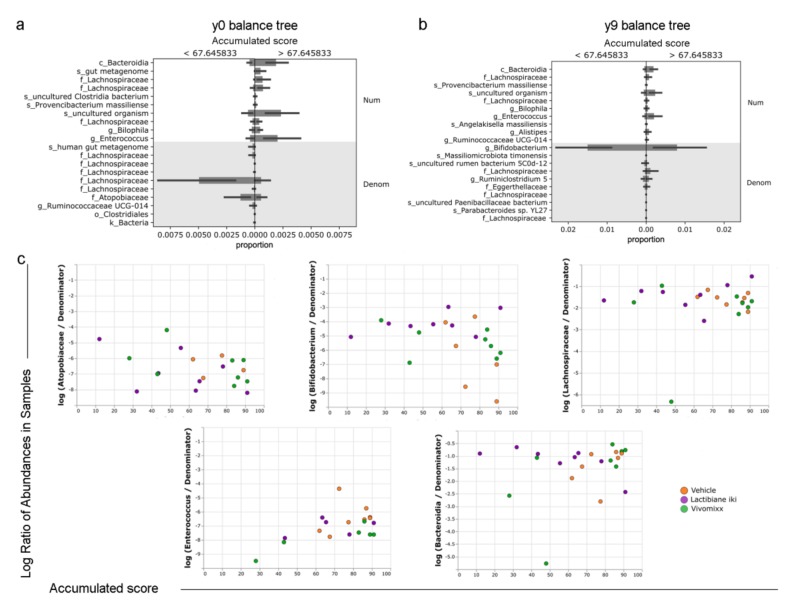
Specific bacterial taxa are associated with disease progression. Faeces were freshly collected in duplicate from every treated experimental autoimmune encephalomyelitis (EAE) mouse (n = 8 per group of treatment: vehicle, Lactibiane iki, or Vivomixx) at 33 days postimmunization (dpi). After collection, samples were frozen by immersion in liquid nitrogen and stored at −80 °C. Once 16S rDNA sequencing and quality controls were performed, alpha and beta diversity studies were performed. Disease progression was previously associated with differential taxon abundance values with reference to the value distribution of the continuous variable accumulated score (the response variable of interest) as analysed by the Gneiss method. The Gneiss method constructs taxon balance trees (for example, balance y0 (**a**) and y9 (**b**)) and performs multivariate response linear regression to assess whether these balance trees show statistically significant differences along the value distribution of the response variable of interest. Class *Bacteroidia* and genus *Enterococcus* were overrepresented in the numerator of both the y0 and y9 balance trees (related to a higher accumulated score in mice), whereas the families *Lachnospiraceae* and *Atopobiaceae* and the genus *Bifidobacterium* were highly represented in the denominator of the y0 and y9 balance trees (related to a lower accumulated score in mice), respectively (**a**,**b**). The log ratios of abundance values for selected taxa within samples were plotted using the q2-deicode and q2-qurro plugins of QIIME2 (**c**). The abundance of both the family *Atopobiaceae* and the genus *Bifidobacterium* relative to the overall microbial community is higher in mice with a lower accumulated score. However, no such tendency was observed within the *Lachnospiraceae* family. Visual inspection of *Enterococcus* revealed a higher abundance of this bacterial genus in mice with a higher accumulated score, while no differences were observed regarding the class *Bacteroidia*. The graphs show the results of a representative experiment under double dose administration (vehicle-treated mice (33 dpi), n = 8; Lactibiane iki-treated mice (33 dpi), n = 8; and Vivomixx-treated mice (33 dpi), n = 8). The data are presented as the means ± standard deviations. Abbreviations: Denom: denominator; Num: numerator.

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
