# Peer review of "A Commercial Probiotic Induces Tolerogenic and Reduces Pathogenic Responses in Experimental Autoimmune Encephalomyelitis"

_cells, 2020, doi:10.3390/cells9040906_

Round 1
Reviewer 1 Report
It is a very interesting and well-performed study. The aims of the study are well-defined and the experimental methodology is described in detail. The results are presented nicely and discussed sufficiently. My only comment is on the introduction. Lines 79-89: ‘’In our EAE mice….it is already a commercialized product’’. The authors should move this paragraph to the conclusions and modify the introduction and conclusions accordingly.
Author Response
It is a very interesting and well-performed study. The aims of the study are well-defined and the experimental methodology is described in detail. The results are presented nicely and discussed sufficiently. My only comment is on the introduction. Lines 79-89: ‘’In our EAE mice….it is already a commercialized product’’. The authors should move this paragraph to the conclusions and modify the introduction and conclusions accordingly.
We thank the reviewer for this comment. The authors have integrated this paragraph in the conclusions section (Lines 697-702) and modify the introduction section (Lines 80-84) accordingly.
Reviewer 2 Report
In this manuscript, Calvo-Barreiero et al assess the effect of two commercially available probiotics in the progression of established EAE. The authors conclude that the probiotic Lactibiane iki alleviates EAE symptoms while this is not the case for Vivomixx, a probiotic that was previously shown promise in MS patients.
The authors show that Lactibiane iki impairs T cells activation in the periphery, promotes Treg generation and induces PD-L1 expression in the mDCs; these result in impaired infiltration of the spinal cord by T cells thus leading to diminished axonal damage. This is a nicely written manuscript with a lot of information that may prove helpful for the design of MS treatments.
Comments:
Figure 1 shows that both Vivomixx and Lactibiane iki improved the rotatord score compared to the vehicle treated mice, however Vivomixx-treated mice did not show statistically significant clinical protection in EAE compared to the controls. Could the authors explain why this was the case since EAE is scoring is mainly based on motor movement?
The authors show that doubling the dose of treatment (2x/day) further improves the EAE clinical course in the Lactibiane iki group and although the Vivomixx group shows some improvement this again does not reach statistical significant differences compared with the vehicle treated controls. Which regimen did the authors employ for the rest of the figures shown in the paper?
Figure 2 shows that both treatment groups show similarly decreased levels of demyelination and of CD3+ T cells present in the spinal cord on day 34. At the same time point the authors show that there are fewer SMI32+ axons in the Lactibiane iki group compared to vehicle control. Interestingly the numbers of SMI32+ axons in Vivomixx group are not statistically significantly different than the vehicle controls. Although this difference may be attributed to lower Th17 cell number in the CNS of Lactibiane iki compared to Vivomixx and vehicle groups (as shown in figure 3), I wonder whether this could also be due to antecedent myeloid cell infiltration, which is also associated with axonal damage. By day 34 the inflammatory environment in the CNS is usually much less intense compared to that in disease onset and/or peak of disease and the authors might have missed differences in myeloid cells in the CNS between the groups. They could however discuss this possibility. On that note the authors mention that there were no differences between the groups in microglial and astrocyte reactivity; how did the authors examine these parameters? Did they try analysis by fluorescence intensity?
Did the authors examine monocytic/neutrophilic cell subsets? Although a number of myeloid cell markers are mentioned in the material and method section, there are no such data presented in the results. It would have been very informative to see whether myeloid subsets (other than DCs) are affected in the periphery or the CNS with this treatment.
In figure 5 representative flow dot plots for PD-L1 and CD86 in mDCs and MHCII and CD80 MFIs would be helpful.
Were the data shown in the figures collected from one experiment or multiple experiments pooled together?
Minor comment: I would not call IL-4 anti-inflammatory. It is an inhibitor of Th17 and Th1 cell polarization but it is an inflammatory cytokine in allergic and atopic environment.
Author Response
In this manuscript, Calvo-Barreiero et al assess the effect of two commercially available probiotics in the progression of established EAE. The authors conclude that the probiotic Lactibiane iki alleviates EAE symptoms while this is not the case for Vivomixx, a probiotic that was previously shown promise in MS patients.
The authors show that Lactibiane iki impairs T cells activation in the periphery, promotes Treg generation and induces PD-L1 expression in the mDCs; these result in impaired infiltration of the spinal cord by T cells thus leading to diminished axonal damage. This is a nicely written manuscript with a lot of information that may prove helpful for the design of MS treatments.
We thank the reviewer for these remarks.
Comments:
Figure 1 shows that both Vivomixx and Lactibiane iki improved the rotatord score compared to the vehicle treated mice, however Vivomixx-treated mice did not show statistically significant clinical protection in EAE compared to the controls. Could the authors explain why this was the case since EAE is scoring is mainly based on motor movement?
We thank the reviewer for this comment. As the reviewer has mentioned, there is no statistically significant difference regarding EAE score between Vivomixx and vehicle group even though there are significant differences on the Rotarod test. Although Rotarod apparatus tests motor coordination skills, EAE scoring is a more sensitive approach to measure motor abilities since it takes into account: mouse motor coordination skills, mouse capacity to grip and walk on a straight and a steep trellis, and mouse strength to hold onto a trellis. In addition, the clinical score (area under the curve) is a daily measure while the Rotarod test was performed at the end of the experiment when the differences between groups were maximum. For these reasons, the authors explain this discrepancy between these two supplementary ways of motor assessment.
The authors show that doubling the dose of treatment (2x/day) further improves the EAE clinical course in the Lactibiane iki group and although the Vivomixx group shows some improvement this again does not reach statistical significant differences compared with the vehicle treated controls. Which regimen did the authors employ for the rest of the figures shown in the paper?
The authors had already detailed which regimen has been employed for every studied parameter in every figure caption included in the article.
Figure 2 shows that both treatment groups show similarly decreased levels of demyelination and of CD3+ T cells present in the spinal cord on day 34. At the same time point the authors show that there are fewer SMI32+ axons in the Lactibiane iki group compared to vehicle control. Interestingly the numbers of SMI32+ axons in Vivomixx group are not statistically significantly different than the vehicle controls. Although this difference may be attributed to lower Th17 cell number in the CNS of Lactibiane iki compared to Vivomixx and vehicle groups (as shown in figure 3), I wonder whether this could also be due to antecedent myeloid cell infiltration, which is also associated with axonal damage. By day 34 the inflammatory environment in the CNS is usually much less intense compared to that in disease onset and/or peak of disease and the authors might have missed differences in myeloid cells in the CNS between the groups. They could however discuss this possibility. On that note the authors mention that there were no differences between the groups in microglial and astrocyte reactivity; how did the authors examine these parameters? Did they try analysis by fluorescence intensity?
We are grateful for these excellent comments. The authors agree that the CNS inflammatory environment is lower at 34 days post immunization (chronic phase) than at disease onset or at the peak of the disease. However, since experimental treatments started to be administered once the EAE was stablished, the authors did not have the opportunity to investigate treatments’ effects on myeloid cells in these early time points.
On the other hand, reactive microglia/macrophages and astroglia were assessed by lectin from Lycopersicon esculentum and anti-glial fibrillary acidic protein, respectively. Although used by several authors, the measurement of the intensity of immunostaining is not a reliable way to calculate cell activity. The histological procedure, even if carried out by the most experienced hands, can be a source of variability in the intensity of each immunostaining and can be very tricky for some antibodies. To avoid this biological/quantitative error, we decided to quantify microglia and astrocyte activation as the area with specific staining relative to the total white matter area. The results were shown as the percentage of positive area relative to the total white matter area. An inflammatory insult like EAE produces an extensive infiltration of monocyte-derived cells as well as a proliferative response in the resident microglial cell population (Lassmann H. Pathology of inflammatory diseases of the nervous system: Human disease versus animal models. Glia. 2020 Apr;68(4):830-844). Thus, a more inflamed CNS is followed by an increase in the number of stained cells. For this reason, the quantification of the stained area is enough to measure the activation or probiotic-mediated deactivation state of microglia/macrophages in a morphological context like the CNS of EAE mice. Regarding astrocytes, EAE induces the hypertrophy and branching of these macroglial cells (Tani M, Glabinski AR, Tuohy VK, Stoler MH, Estes ML, Ransohoff RM. In situ hybridization analysis of glial fibrillary acidic protein mRNA reveals evidence of biphasic astrocyte activation during acute experimental autoimmune encephalomyelitis. Am J Pathol. 1996 Mar;148(3):889-96) which means that a more activated astrocyte has a wider area within the CNS. For this reason, our quantitative strategy for astrocyte activation/inactivation allows us to measure their activity state and the effect of probiotics on this specific glial cell population.
Did the authors examine monocytic/neutrophilic cell subsets? Although a number of myeloid cell markers are mentioned in the material and method section, there are no such data presented in the results. It would have been very informative to see whether myeloid subsets (other than DCs) are affected in the periphery or the CNS with this treatment.
We thank the reviewer for this excellent comment. Besides already analysed macrophage and dendritic cell populations, the authors have considered this comment and they have also analysed myeloid-derived suppressor cell (MDSC) populations in the periphery as follows:
|
Myeloid-derived suppressor cells (MDSCs) |
|
|
Monocytic MDSCs |
Ly6G- CD11b+ Ly6Chigh |
|
Granulocytic MDSCs |
Ly6G+ CD11b+ Ly6Clow |
Moliné-Velázquez V, Vila-Del Sol V, de Castro F, Clemente D. Myeloid cell distribution and activity in multiple sclerosis. Histol Histopathol. 2016 Apr;31(4):357-70.
Whereas no differences were observed regarding monocytic MDSCs between experimental groups, Lactibiane iki treatment reduced the granulocytic MDSCs in the periphery. Unfortunately, with the available cell markers included in the flow cytometry panel (CD11b, Ly6G, Ly6C, F4/80 and CD206), we cannot distinguish between mature neutrophils and granulocytic MDSCs since these two immune cell populations share morphology and cell surface markers. Thus, immunosuppression in vitro assays would be necessary for this immune cell characterisation, but this study is beyond the scope of this paper.
Casacuberta-Serra, S, Parés, M, Golbano, A, Coves, E, Espejo, C. & Barquinero, J. Myeloid-derived suppressor cells can be efficiently generated from human hematopoietic progenitors and peripheral blood monocytes. Immunology and Cell Biology. 2017;95(6), 538–548.
Casacuberta-Serra S, Costa C, Eixarch H, Mansilla MJ, López-Estévez S, Martorell L, Parés M, Montalban X, Espejo C, Barquinero J. Myeloid-derived suppressor cells expressing a self-antigen ameliorate experimental autoimmune encephalomyelitis. Exp Neurol. 2016 Dec; 286:50-60.
For all these reasons, we will refer to this cell population as neutrophils/granulocytic MDSCs. We have modified Materials & Methods (Line 191), Results (Lines 464-468, 477), Figure 5 (Line 469), Discussion (Lines 646-649) and References (Lines 918-919) sections in order to detail that these quantifications were performed and the obtained results.
In figure 5 representative flow dot plots for PD-L1 and CD86 in mDCs and MHCII and CD80 MFIs would be helpful.
We thank the reviewer for this comment. The authors have modified Figure 5 (Line 469) and the Figure 5 caption (Lines 470-482) as the Reviewer #2 has suggested.
Were the data shown in the figures collected from one experiment or multiple experiments pooled together?
The data shown in the figures of the article were collected from multiple experiments pooled together. The authors had already detailed which regimen and number of pooled experiments has been employed for every studied parameter in every figure caption included in the article.
Minor comment: I would not call IL-4 anti-inflammatory. It is an inhibitor of Th17 and Th1 cell polarization but it is an inflammatory cytokine in allergic and atopic environment.
We appreciate this comment from the reviewer. Although the authors agree with the comment of the Reviewer #2 regarding inflammatory properties of IL-4, the authors think that, in the context of EAE model and MS patients, IL-4 has been widely described as an anti-inflammatory molecule. In fact, protection from experimental autoimmune encephalomyelitis has been achieved with IL-4 therapy and IL-4 deficient mice developed a more severe form of clinical disease:
Furlan R, Poliani PL, Galbiati F, Bergami A, Grimaldi LM, Comi G, Adorini L, Martino G. Central nervous system delivery of interleukin 4 by a nonreplicative herpes simplex type 1 viral vector ameliorates autoimmune demyelination. Hum Gene Ther. 1998 Nov 20;9(17):2605-17
Falcone M, Rajan AJ, Bloom BR, Brosnan CF. A critical role for IL-4 in regulating disease severity in experimental allergic encephalomyelitis as demonstrated in IL-4-deficient C57BL/6 mice and BALB/c mice. J Immunol. 1998 May 15;160(10):4822-30)